# Can MDCT Enhancement Patterns Be Helpful in Differentiating Secretory from Non-Functional Adrenal Adenoma?

**DOI:** 10.3390/medicina60010072

**Published:** 2023-12-29

**Authors:** Svetlana Kocic, Vladimir Vukomanovic, Aleksandar Djukic, Jovica Saponjski, Dusan Saponjski, Vuk Aleksic, Vesna Ignjatovic, Katarina Vuleta Nedic, Vladan Markovic, Radisa Vojinovic

**Affiliations:** 1Department of Radiology, Clinical Hospital Center Zemun, 11070 Belgrade, Serbia; lanakocic@gmail.com; 2Department of Nuclear Medicine, Faculty of Medical Science, University of Kragujevac, 34000 Kragujevac, Serbia; vesnaivladaignjatovic@gmail.com (V.I.); kvuleta87@gmail.com (K.V.N.); 3University Clinical Center Kragujevac, 34000 Kragujevac, Serbia; adjukic@sbb.rs (A.D.); drjack.vm@gmail.com (V.M.); rhvojinovic@medf.kg.ac.rs (R.V.); 4Department of Pathophysiology, Faculty of Medical Science, University of Kragujevac, 34000 Kragujevac, Serbia; 5Faculty of Medicine, University of Belgrade, 11000 Belgrade, Serbia; sapica1961@yahoo.com (J.S.); saponjski.d@gmail.com (D.S.); 6University Clinical Center of Serbia, 11000 Belgrade, Serbia; 7Department of Neurosurgery, Clinical Hospital Center Zemun, 11070 Belgrade, Serbia; aleksicvuk@hotmail.com; 8Department of Radiology, Faculty of Medical Science, University of Kragujevac, 34000 Kragujevac, Serbia

**Keywords:** multidetector computed tomography, scintigraphy, adrenal tumor, ^99m^Tc-HYNIC-TOC

## Abstract

*Background and Objectives:* Primary adrenal tumors (AT) are a heterogeneous group of neoplasms due to their functional heterogeneity, which results in the diverse clinical presentation of these tumors. The purpose of this study was to examine cross-sectional imaging characteristics using multi-detector computed tomography (MDCT) to provide insight into the lesion characterization and functional status of these tumors. The radionuclide imaging using Technetium-99m radiolabeled hydrazinonicotinylacid-d-phenylalanyl^1^-tyrosine3-octreotide (^99m^Tc-HYNIC-TOC), was also used in the diagnostic evaluation of these tumors. *Materials and Methods*: This cross-sectional study included 50 patients with confirmed diagnoses of AT (21 hormone-secreting and 29 non-functional) at the University Clinical Center, Kragujevac, Serbia, during the 2019–2022 year period. The morphological and dynamic characteristics using MDCT were performed, using qualitative, semi-quantitative, and quantitative analysis. Absolute washout (APW) and relative washout (RPW) values were also calculated. A semi-quantitative analysis of all visual findings with ^99m^Tc-HYNIC-TOC was performed to compare the tumor to non-tumor tracer uptake. *Results*: A statistically significant difference was found in the MDCT values in the native phase (*p* < 0.05), the venous phase (*p* < 0.05), and the delayed phase (*p* < 0.001) to detect the existence of adrenal tumors. Most of these functional adrenocortical lesions (n = 44) can be differentiated using the delayed phase (*p* < 0.05), absolute percentage washout (APW) (*p* < 0.05), and relative percentage washout (RPW) (*p* < 0.001). Furthermore, ^99m^Tc-HYNIC-TOC could have a high diagnostic yield to detect adrenal tumor existence (*p* < 0.001). There is a positive correlation between radionuclide imaging scan and APW to detect all AT (*p* < 0.01) and adrenocortical adenomas as well (*p* < 0.01). *Conclusions:* The results can be very helpful in a diagnostic algorithm to quickly and precisely diagnose the expansive processes of the adrenal glands, as well as to learn about the advantages and limitations of the mentioned imaging modalities.

## 1. Introduction

Benign or malignant primary adrenal tumors (AT) are a heterogeneous group of neoplasms characterized by a marked diversity of the clinical presentation, which results from the functional, structural, embryological, and evolutionary heterogeneity of normal endocrine cells and it is associated with high rates of morbidity and mortality [1]. While the majority of morphological disorders are benign tumors (adenomas), some of them behave like truly malignant tumors with the capacity for metastatic spread and the possibility of fatal evolution [1,2]. Adrenal incidentalomas (AI) are often asymptomatic masses that are usually detected using imaging diagnostic techniques performed for other purposes [2,3]. Clinical manifestations of functionally active tumors depend on specific hormone secretion, although plurihormonal secretion is the most common [2,3,4]. Although most AI are benign, non-functional tumors, 10% of these masses are related to abnormal hormone secretion and are rarely found to be malignant [4,5,6]. Thus, it is essential to distinguish functional from non-functional AT. Increased incidence and prevalence of AT are dominant in older people, as well as in female gender [5,6,7]. According to recent studies, up to 80% of AT are non-functional adenomas, 20–30% secrete cortisol, and up to 10% are aldosterone-secreting tumors [6,7,8]. The prevalence of pheochromocytomas is 3–6%, while androgen- or estrogen-secreting masses, primary malignancies, and adrenal metastases are extremely rare [8,9]. Due to the specific localization and diversity of the clinical presentation, the expansive processes of the adrenal glands represent a diagnostic challenge. Available diagnostic procedures should have high specificity and sensitivity to avoid a high rate of false negative findings [9,10]. According to current guidelines, the European Society of Endocrinology, American College of Radiology, Korean Endocrine Society, American Association of Clinical Endocrinologists and American Association of Endocrine Surgeons, and Canadian Urological Association, the optimal evaluation of adrenal masses larger than 1 cm requires a multidisciplinary team approach including clinical, radiologic, and biochemical assessment. Although they are very well designed, the lack of large long-term studies has led to multiple variations [1,11,12,13,14].

The most significant points of divergence are imaging and hormonal follow-up protocol, management of indeterminate adrenal masses that are not characterized as benign or malignant, indications for biopsy, and surgical treatment. For non-functioning adrenal mass, with HU <10 on native CT, there is still no consensus for a period of imaging and hormonal follow-up. For the non-functioning adrenal mass, with HU values from 11 to 20 HU on native CT, all guidelines recommend imaging follow-up, but with no consensus on interval, highlighting the need for further studies to identify dedicated imaging protocols to be incorporated into the decision-making process. Indeterminate adrenal masses should have an individualized patient approach in a multidisciplinary team meeting, with a need for updates that consider individual risk evaluation, size, homogeneity, and HU values of adrenal masses on CT imaging. Management protocol for cortisol-secretory adenoma supports that adrenalectomy should be considered based on an individual patient’s clinical condition. According to all guidelines, pheochromocytoma must be resected with a long-term imaging follow-up for possible recurrence, for aldosterone-secreting adenomas adrenalectomy or medicament treatment is also based on an individual patient approach [13]. An invasive procedure, biopsy should not be a part of an initial workup, again emphasizing the need for upgrading the use of non-invasive diagnostic modalities. Surgery is advisable for adrenal masses larger than 4 cm, because of the high malignancy risk, but for smaller lesions, there is still no consensus. MDCT examination is the recommended diagnostic modality, while MRI should be used selectively and as the first choice for children, pregnant women, breastfeeding women, and patients with hypersensitivity to iodinated contrast agents, impaired renal function, and fear of ionizing radiation [1,11,12,13,14].

According to comparative radiology studies, radionuclide imaging has a complementary and increasingly important role in characterizing adrenal lesions [15].

Radiolabeled isotope studies serve as second-line diagnostic tests for malignant adrenal tumors, primary or metastatic, as well as for pheochromocytoma. Scintigraphy with the somatostatin analog tyrosine3-octreotide labeled with the radioactive isotope Tc-99m (^99m^Tc-HYNIC-TOC) has an affinity for tumors expressing somatostatin receptors. ^99m^Tc-HYNIC-TOC has high sensitivity and specificity and enables precise localization of expansive changes in the adrenal glands [15].

Considering the significant points of divergence among current guidelines, the difficulty of performing hormonal tests, and the growing precision of imaging modalities, there is a real need for a dedicated imaging protocol. Therefore, this study aims to investigate MDCT imaging characteristics of adrenal masses to differentiate secretory from non-functioning adenomas.

## 2. Material and Methods

### 2.1. Study Population

This retrospective, clinical, non-invasive, cross-sectional study included 50 patients with a confirmed diagnosis of AT based on biochemical results. The control group presented a contralateral healthy adrenal gland (n = 50). The study was conducted during the 2019–2022 year period at the University Clinical Center, Kragujevac, Serbia, in accordance with the Declaration of Helsinki (World Medical Association) and an institutional ethical committee under official number 01/22.199. Informed consent has been obtained for all subjects before the diagnostic procedures. Exclusion criteria were: age < 18 years, pregnancy, breastfeeding, diseases or drugs influencing hormonal secretion, disorders with a similar clinical presentation, systemic or infiltrative diseases potentially affecting the adrenal glands, history of malignant disease, and other severe life-threatening diseases, and no consent given.

### 2.2. Biochemical Parameters

All patients underwent a standardized diagnostic evaluation based on biochemical, clinical parameters, and imaging criteria. The functional status of adrenal glands was evaluated by baseline hormonal assessment, which included serum cortisol (with overnight, low-dose, and high-dose dexamethasone suppression test-DST), adrenocorticotropic hormone (ACTH), prolactin, potassium, aldosterone, progesterone, testosterone and β-estradiol, luteinizing hormone (LH) and follicle-stimulating hormone (FSH), and plasma-free metanephrine and catecholamines.

### 2.3. MDCT Imaging Protocol

The imaging evaluation of adrenal glands was assessed with a 64-row multi-detector CT scanner (Aquilion™, Toshiba, Otawara, Japan) slice thickness 0.5 mm, and increment 0.5 mm, rotation time 0.5 s; 120/200 kVp/mAs with an automatic dose modulation system. The dedicated CT examination protocol consisted of pre-contrast and post-contrast scanning with an iodinated contrast agent, in native, arterial (25 s), early wash-in or venous (60 s), delayed phase (15 min), and percentage quantification of absolute and relative wash out. The images were evaluated by two experienced radiologists separately, unaware of the clinical and laboratory results of the patients, using a workstation (Advantage Workstation Toshiba, Otawara, Japan). Interobserver agreement on CT measurements using an Intraclass Correlation Coefficient (ICC) was excellent, for native CT phase ICC = 1.000, 95%CI (0.999–1.000), arterial phase ICC = 0.999, 95%CI (0.999–1.000), venous phase ICC = 0.998, 95%CI (0.995–0.999), delayed phase ICC = 0.997, 95%CI (0.995–0.999), RPW ICC = 0.990, 95%CI (0.982–0.995), APW ICC = 0.995, 95%CI (0.988–0.996). We evaluated the native, arterial, venous phase, and delayed phase. This study analyzed morphological features such as size, shape, border, side of the expansion, lipid component, and homogeneous or heterogeneous density before and after contrast administration. A semi-quantitative analysis of all visual findings of cystic, necrotic, calcified, or hemorrhagic areas was evaluated. In the axial plane, CT density values (HU units) were measured in all four phases of CT examination. A circular region of interest (ROI) was used to measure CT attenuation. The ROI was placed in all phases in a central region covering 1/2 to 2/3 of the mass away from the periphery to prevent a partial volume effect. Cystic, necrotic, calcified, or hemorrhagic areas were not included in the ROI since they could affect the results (Figure 1).

Absolute washout (APW) and relative washout (RPW) values were also calculated using the defined formulas based on attenuation values in HU units on native, unenhanced CT phase (UCT), enhanced venous phase (ECT), and delayed phase (DCT) as follows: APW = [ECT (HU) − DCT (HU)] × 100/[ECT(HU) − UCT(HU)] and RPW = [ECT (HU) − DCT (HU)] × 100/ECT (HU).

### 2.4. Nuclear Medicine Imaging Protocol

All patients underwent radionuclide imaging with ^99m^Tc-HYNIC-TOC. According to the manufacturer’s instructions, the tracer was prepared using a commercially available kit, with recommended administered activity. The acquisition protocol consisted of whole-body scans in the anterior and posterior projections (256 × 1024 matrix, 12 cm/min), performed on dual head Gamma camera (Syngo-E-cam™, Siemens, Berlin, Germany), with low energy high-resolution collimators (window 140 keV). A single-photon emission computed tomography (SPECT) scan of the region of adrenal glands was conducted with the following parameters: 360° non-circular orbit (body contour mode) step and shoot mode, at 30 s per view, 1.23 zoom. The obtained data firstly were collected in a 128 × 128 image matrix, and then reconstructed in all projections using an iterative ordered subset expectation (OSEM) algorithm. All images were interpreted qualitatively and semi-quantitatively by two experienced nuclear medicine physicians unaware of the clinical and laboratory results of the patients (Figure 1). Interobserver agreement on measurements was excellent ICC = 1.000, 95%CI (0.999–1.000). The tumor to non-tumor tracer uptake was calculated using the total counts/total pixels ROI technique, from the tumor’s ROI to the contralateral mirror ROI, respectively. All data were analyzed using a Syngo-E-cam™ system.

### 2.5. Statistical Analysis

Data collected through clinical research were organized into a Microsoft Excel file from where they were exported to the statistical program IBM SPSS2C3 (Chicago, IL, USA), where they were adjusted and with the help of statistical analyses from the measures of descriptive statistics, frequencies, and percentages, and the arithmetic was used mean, standard deviation, median, and minimum and maximum value. For analytical statistics methods, we used: the Chi-Square test, Fisher’s exact probability test, Student’s *t*-test, analysis of variance (ANOVA), Mann–Whitney U-test, and receiver operating characteristic curves (ROC). The alpha level for significance was set to *p* < 0.05. The intraclass correlation coefficient (ICC) was calculated to evaluate interobserver agreement.

## 3. Results

The study involved 50 consecutive patients, 16 male (32.0%), and 34 female (68.0%). The average age of the patients was 59.12. The youngest patient was 33 years old and the oldest was 76 years old. Although gender distribution among patients with ATs varies in different series, females are most commonly affected predominantly in the fourth to sixth decade of life. In the present study gender distribution of adrenal masses is by the majority of investigations [5,6,7]. The masses were detected more frequently in the left adrenal gland 28 (56.0%) compared to the right 22 (44%). The mean diameter of the adrenal mass was (30.06 mm, SD 14.60), ranging from 10 mm to 72 mm.

The results of the Wilcoxon Signed Ranks Test of the mean values by HU stages between AT and control groups showed statistically significant differences in the native phase (Z = −2.462, *p* < 0.05), the venous phase (Z = −2.800, *p* < 0.05), and the delayed phase (Z = −3.664, *p* < 0.001). Furthermore, the ^99m^Tc-HYNIC-TOC target/non-target ratio scan showed a highly statistically significant difference in the mean values T/NT ratios in the AT group and controls, respectively (Z = −6.154, *p* < 0.001) (Table 1).

### Grouping Criteria for Patients

All patients were divided into two groups based on biochemical results: secretory adrenal masses 21 (42%), which included cortical adenomas and pheochromocytomas (30% and 12%, respectively), and non-secretory masses 29 (58%). We examined morphological and dynamic characteristics using MDCT to provide insight into lesion characterization and functional status of these tumors. Appendix A. MDCT morphological characteristics of adrenal masses obtained in the unenhanced phase in relation to secretory activity.

The attenuation values of the adrenal masses on native phase and enhanced phases (arterial, venous, and delayed) CT imaging, absolute and relative washout, and ^99m^Tc-HYNIC-TOC target/non-target ratio are shown in Table 2.

The Mann–Whitney U-test for these two groups showed a statistically significant difference in the values of relative washout (*p* = 0.001) and absolute washout (*p* < 0.001). Patients with secretory tumors of the adrenal gland had higher mean values than patients with non-functioning tumors (Table 3).

Furthermore, we excluded patients with pheochromocytomas (n = 6), and patients with adrenal cortical adenoma (n = 44) were divided into two groups: secretory adenoma (n = 15) and non-functional adenoma (n = 29). Among secretory adenomas, 10 (20%) were cortisol-secreting, 5 (10%) were aldosterone-secreting, and none androgen-secreting. Appendix A. MDCT morphological characteristics between secretory and nonfunctional adenoma obtained in the unenhanced phase in relation to secretory activity.

Patients with non-secretory adenoma had an average age of 58.26 years, and patients with secretory adenoma were 60.53 with no statistically significant difference (*p* = 0.444). The mean diameter of secretory masses was (37.73, SD 20.20), non-functional (26.86, SD 10.49) with no statistically significant difference (*p* = 0.154). The masses with dimensions greater than 4 cm were present in secretory adenoma 5 (n = 15) and non–non-functioning adenoma 3 (n = 29). The dominant lipid component was present in non-functional adenomas 20 (n = 29) and 7 (n = 15) in secretory adenomas, with the mean native CT value of secretory (10.4 mm, SD 19.42), and non-functional adenomas (13.24 mm, SD 17.89) with no statistically significant difference (*p* = 0.594). Both groups of adenomas had smooth borders without cystic, necrotic, or hemorrhagic areas. Area of calcifications were present in the group of secretory adenoma 3 (n = 15), whose dimensions were greater than 4 cm (n = 5). Area of necrosis was present only in the group of pheochromocytomas 3 (n = 6).

The MDCT attenuation values of the adrenal cortical-secretory and non-functional adenoma on native arterial, venous, and delayed contrast-enhanced imaging, absolute and relative wash out and ^99m^Tc-HYNIC-TOC target/non-target ratio are shown in Table 4.

The Mann–Whitney U-test showed a statistically significant difference between secretory and non-functional adenomas for delayed phase (*p* = 0.027), relative washout (*p* = 0.001), absolute washout (*p* = 0.00) and ^99m^Tc-HYNIC-TOC target/non-target (*p* = 0.046) (Table 5). Patients with secretory tumors have higher values than patients with non-functional adenomas, while the opposite is true for the delayed phase, patients with secretory tumors have significantly lower values than patients with non-functional (Table 5).

The ROC curve for RPW and secretory activity, with area under curve (AUC) 0.813, 95%CI (0.672–0.953), was statistically highly significant (*p* < 0.001), and the cut-off value was 47.5% and above, with a sensitivity of 73.3%, 95%CI (44.9–92.0%), and the specificity was 86.2%, 95%CI (68.3–96.0%).

The ROC curve for APW and secretory activity, with AUC 0.989, 95%CI (0.963–1.000), was statistically highly significant (*p* < 0.001), the cut-off value was 59.5% and above, with a sensitivity of 93.3%, 95%CI (68.0–98.9%), and the specificity was 100.0%, 95%CI (87.9–100.0%).

The ROC curve for delayed CT and secretory activity, with AUC 0.705, 95%CI (0.548–0.861), was statistically significant (*p* = 0.011), and the cut-off value was 47.5 HU and under, with a sensitivity of 80%, 95%CI (51.9–95.4%), and the specificity was 55.9%, 95%CI (35.7–73.5%).

The ROC curve for RPW and lesion size, with AUC 0.681, 95%CI (0.505–0.857), was statistically significant (*p* =0.0441), the cut-off value was 43.5% and above, with a sensitivity of 77.8%, 95%CI (40.1–96.5%), and the specificity was 54.3%, 95%CI (36.7–71.2%).

The ROC curve for APW and lesion size, with AUC 0.743, 95%CI (0.574–0.912), was statistically significant (*p* = 0.005), the cut-off value was 56.5% and above, with a sensitivity of 77.8%, 95%CI (40.1–96.5%), and the specificity was 65.7%, 95%CI (47.8–80.9%). The ROC curve for delayed CT phase and lesion size, with AUC 0.551, 95%CI (0.318–0.783), was not statistically significant (*p* = 0.668), and the cutoff values could not be determined.

When analyzing the correlation between two imaging modalities’ scintigraphy and MDCT parameters in the group of all Ats (n = 50, secretory/non-functional adenoma and pheochromocytoma), there is a statistically significant positive correlation between the variable ^99m^Tc-HYNIC-TOC target/non-target ratio and the variable APW (r = 0.362; *p* = 0.010). There is no statistically significant correlation with other variables. The coefficient of determination R2 = 0.131. ANOVA shows the statistical significance of the model (F = 7.219; *p* = 0.010).

Additionally, when analyzing the same parameters in the cortical adenoma group (n = 44, secretory/non-functional adenoma), there is a statistically significant positive correlation between the variable ^99m^Tc-HYNIC-TOC target/non-target ratio and the variable APW (r = 0.387; *p* = 0.009). There is no statistically significant correlation with other variables. The coefficient of determination R2 = 0.150. ANOVA shows the statistical significance of the model (F = 7.406; *p* = 0.009).

## 4. Discussion

The recommended evaluation of adrenal masses includes a clinical, radiologic, and biochemical assessment with the primary goal of distinguishing benign from malignant tumors, as well as non-functioning from functioning masses [1,11,12,13,14].

From the functional perspective, hypercortisolism, hyperaldosteronism, and catecholamine secretion are the three main secreting syndromes, while androgen secretion is rare [11,12,13,14]. In our study, based on obtained biochemical results, among secretory adenoma, cortisol-secreting were 10 (20%), aldosterone-secreting 5 (10%), and androgen-secreting tumors have not been detected, which is the same as a majority of reported studies [6,7,8]. Secretions can be clinically significant or may be subtle and associated only with biochemical abnormalities. Hormonal evaluation of all incidentalomas can be very costly, so they are suggested only “if there are evident clinical signs or symptoms”, which is contrary to the endocrine literature where “adrenal secretion excess may be present also without clinical symptoms”. In fact, in up to 30% of patients, a mild increase in cortisol secretion was detected without any clinical signs [2,3,4,5,6,16]. The increasing knowledge that these patients may be at high risk for hypertension, obesity, diabetes, osteoporosis, cardiovascular events, and mortality still presents a source of uncertainties, leading to misclassification of these patients mainly because of the lack of standardized procedures [5,6,16].

Current guidelines recommend MDCT and MRI as imaging techniques for the evaluation of adrenal masses, but still without randomized studies comparing imaging modalities [1,11,12,13,14]. MDCT examination is more frequently used, with a prevalence varying between 0.35% and 9% in different series, while MRI remains a second-level technique. The expense, the duration of examination, which is very demanding for the patient, as well as the appearance of artifacts caused by physiological processes in the abdomen, are cited as the main limiting factors [11,12,13,14,15]. The most important MRI technique is Chemical-shift which can be used to distinguish adenomas from non-adenomas. Studies have shown that for lipid-rich adenomas, there is no difference between MDCT and MRI. However, the accurate differentiation of lipid-poor adenoma from non-adenoma remains a diagnostic challenge. With the use of APW and RPW, CT achieves an accuracy of 98% in identifying lipid-poor adenomas, while a substantial number can be misdiagnosed in MRI chemical shifts. The MRI DWI technique is not a very useful method, because of the significant overlap between adenoma and non-adenoma reported in studies, while MR spectroscopy is promising but needs more studies to be validated [11,12,13,14,17,18].

Most studies have described the value of native and post-contrast MDCT densitometry in differentiating adrenal adenomas from non-adenomas, especially metastases [10,17,18]. However, up to now, only a few studies have explored whether increased hormonal adenoma secretion, mainly those associated with hypercortisolemia, can be presented with different native and enhanced attenuation on MDCT [19,20]. Monsoni et al. reported that, in patients with subclinical hypercortisolism, a heterogeneous radiologic CT pattern was present, suggesting that functional and morphologic parameters differed between secreting and non-functional adenomas [21]

In our study, we examined whether specific MDCT parameters can be used to differentiate secretory from non-functioning adenomas, concerning the obtained hormonal findings. To our knowledge, both native and post-contrast MDCT densitometry in differentiating all cortical-secretory from non-functional adenomas have not yet been systematically assessed.

Native CT is a very useful imaging modality, firstly, for the evaluation of the morphological features: the size, shape, borders, homogeneous or inhomogeneous appearance, lipid component, cystic, necrotic, calcified, or hemorrhagic areas. These can be very useful elements for differential diagnosis between adenomas and non-adenomas but always need to be combined with other parameters that are in accordance with the majority of investigations [19,20,21,22]. In our study, the mean diameter of secretory masses was (37.73 mm, SD 20.20), non-functional (26.86 mm, SD 10.49). The finding that secretory adenomas were larger than non-functional adenomas confirms what has been reported by others [19,20,21].

Also, native CT is an established imaging modality for characterizing lipid-rich adenomas, measuring 10 HU or less with a sensitivity and specificity of 71% and 98%, respectively [22,23]. Significant differences in the mean lipid content can distinguish adrenal adenomas from non-adenomas where non-adenomas have a native CT attenuation higher than 10 HU but still a very poor sensitivity of 71%, used to diagnose secretory activity [19,20,21,24]. A French study on a large series of patients with adrenal adenomas, with or without clinical symptoms of Cushing syndrome, evaluated the relationship between biochemical, imaging parameters, and histologic characteristics showing that only a minority had an unenhanced attenuation value of less than 10 HU with a poor correlation between this parameter and the lipid content of the adenomas, emphasizing that cortisol hypersecretion might be associated with a variable pattern of histologic and radiologic characteristics [25]. Our study, also showed that both lipid-rich and lipid-poor adenomas can express secretory activity, the dominant lipid component was present in most non-functional adenomas 20 (n = 29) and in 7 (n = 15) secretory adenomas, with the mean native CT value of secretory (10.4 mm, SD 19.42), and non-functional adenoma (13.24 mm, SD 17.89). There was no significant difference between groups (*p* = 0.594). Therefore, in our study, based on native HU values, the secretory activity of adenoma cannot be determined. Only one study in a small series of patients reported that native HU value was the most significant radiological parameter in predicting the functionality of adenoma [26].

Lipid-poor adenomas are defined by native HU values higher than 10 HU and are especially important because they cannot be directly characterized as adenomas or non-adenomas based on native CT attenuation values [27,28,29,30]. Post-contrast phases of CT examination are very important for further dynamic evaluation of adrenal masses. Venous phase (60 s) according to studies does not provide any important element for differential diagnosis, when comparing adenomas and non-adenomas, showing a significant enhancement, with substantially overlapped density values [28,30,31,32,33]. In our study, in the venous phase, we obtained similar results, between non-functional (85.97 HU, SD 27.66) and secretory adenoma (84.47 HU, SD 24.51) with no statistically significant difference (*p* = 0.960). Therefore, a delayed image series is required for further evaluation [29,30,31,32,33,34].

Researchers compared different delay scan times at 5, 10, 15, 30, and 45 min to achieve a simplification of CT scanning time. They concluded that adenomas exhibit significantly more rapid wash out, compared to non-adenomas, which was already evident at 5′ but suggested the delay scan at 15 min, as it was associated with higher sensitivity and specificity for differential diagnosis (88–96% for a 60% APW and 96–100% for a 40% RPW) [35,36,37,38]. Kristin et al. first emphasized that adenomas and non-adenomas could be differentiated based on the lesion wash-out values (APW, RPW) highlighting a more rapid wash-out of adenomas compared with non-adenomas (pheochromocytomas and malignancies), which retain the contrast agents for a longer period [36]. In our study, the delayed scan time was 15 min, which is in accordance with the majority of studies [35,36,37,38,39].

The patients with secretory adenomas had higher values for relative washout (53.47%, SD 10.34), absolute washout (64.80%, SD 7.35), and ^99m^Tc-HYNIC-TOCTU target/nontarget_ratio (25.70, SD 16.89) than patients with non-secretory adenomas relative washout (41.14%, SD 9.25%), absolute washout (48.07%, SD 7.2), and ^99m^Tc-HYNIC-TOC TU target/nontarget ratio (16.06, SD 6.55), while the opposite is true for the delayed phase where patients with secretory adenomas have significantly lower values (38.20, SD 14.29) than patients with non-secretory (51.28, SD 18.80). There was a statistically significant difference between secretory and non-secretory adenomas for delayed phase (*p* = 0.027), relative washout (*p* = 0.001), absolute washout (*p* = 0.00), and ^99m^Tc-HYNIC-TOC TU target/nontarget ratio (*p* = 0.046) concerning the secretion activity.

In our study, APW was a highly significant predictor of secretory activity (AUC = 0.989 (95%CI (0.963–1.000), *p* < 0.001). This was followed by RPW (AUC = 0.813, 95%CI (0.672–0.953), *p* < 0.001) and delayed CT (AUC = 0.705, 95%CI (0.548–0.861), *p* = 0.011). The cut-off for APW 59.5% and above has a sensitivity of 93.3%, 95%CI (68.0–98.9%) and a specificity of 100.0%, 95%CI (87.9–100.0%). The cut-off value for RPW 47.5% and above has a sensitivity of 73.3%, 95%CI (44.9–92.0%) and a specificity of 86.2%, 95%CI (68.3–96.0%). The cut-off value for delayed CT 47.5 HU and under, has a sensitivity of 80%, 95%CI (51.9–95.4%) and a specificity of 55.9%, 95%CI (35.7–73.5%).

Also, APW was a significant predictor of lesion size (AUC= 0.743, 95%CI (0.574–0.912, *p* = 0.005). This was followed by RPW (AUC = 0.681, 95%CI (0.505–0.857), *p* = 0.0441). The area under the ROC was higher for APW and tumor size, and therefore, our results showed that this parameter is a higher predictor. Delayed CT (AUC = 0.551. 95%CI (0.318–0.783, *p* = 0.668 is not a good predictor of lesion size. The cut-off value for APW 56.5% and above has a sensitivity of 77.8%, 95%CI (40.1–96.5%) and a specificity of 65.7%, 95%CI (47.8–80.9%). The cut-off value for RPW43.5% and above has a sensitivity of 77.8%, 95%CI (40.1–96.5%), and a specificity of 54.3%, 95%CI (36.7–71.2%).

This could be explained by the fact that enhancement corresponds to the vascularization of the lesion, so the higher its vascularization, the greater the probability of autonomous secretion [36,37,38,39,40]. These findings confirm that adrenal-secreting adenomas are a heterogeneous group, which is of importance in influencing imaging parameters. One study also reported that enhanced, unenhanced, and delayed attenuation and RPW were significantly different in secretory adenoma in patients with Cushing syndrome [21].

Adrenal scintigraphy using ^99m^Tc-HYNIC-TOC is an SSTR-based imaging method, equal or superior to other conventional imaging methods. Encouraged by the results of various studies somatostatine radiolabeled analog ^99m^Tc-HYNIC-TOC has a high sensitivity in localizing adrenal tumors, while the visualization of functional adrenal masses remains controversial [15,41]. In the present study, we showed statistically significant positive diagnostic potential for detecting the existence of Ats and the functional active form of these tumors as well. The positive correlation between the MDCT absolute washout and ^99m^Tc-HYNIC-TOC is very high (r = 0.387; *p* = 0.009), suggesting that dual imaging methods could have the highest diagnostic yield for AT’s existence.

Our study showed that both lipid-rich and lipid-poor adenoma can express secretory activity, confirming various radiological patterns, but with no significant difference compared to the non-functioning group. In terms of dimensions, all sizes of adrenal adenomas can exhibit hormonal activity. Secretory adenomas were larger than non-functional adenomas but with no significant difference. Native CT attenuation value is not a significant parameter in the detection of secretory adenoma. Delayed CT is a significant predictor of secretory activity. APW and RPW are highly significant predictors of secretory activity.

Although an attenuation value of less than 10 HU on native CT is highly specific for benign lipid-rich adenomas, potential secretory activity cannot be determined based on the unenhanced phase. Given the fact that both lipid-rich and lipid-poor adenoma can express hormone activity, at the initial stage, it is advisable for AI greater than 1 cm to perform dedicated CT protocol with delayed phase and calculate APW and RPW percentage. The obtained data can be of great importance to the clinician in decision-making about further hormonal assessment and follow-up, contributing even more to the reduction of future complications, more effective treatment, as well as better planning of health system resources.

To our knowledge, at present, there is no similar study that examined MDCT characteristics of adrenal adenoma to provide insight into functional status. Only, very few studies investigated CT parameters of secretory adenomas, solely in patients with or without clinical signs and symptoms of Cushing syndrome [21,25].

There are a few limitations to our study. The relatively small sample size of our study, the single-center nature of the study, and the small number of histopathology verifications can be limitations in terms of the use of the functional definition of adrenal adenomas but also support the need for further larger studies on well-selected patients coupled with adequate technological equipment. This kind of study presents the potential to predict the diagnostic capacity of combined endocrinology and imaging parameters, emphasizing the importance of close coordination among professionals from various medical fields.

## 5. Conclusions

Specific CT parameters can be used to provide insight into the functional status of adrenal adenomas in affected patients. APW, RPW, and delayed CT are significant parameters for the prediction of secretory activity, as well as for differentiation between secretory and non-functioning adenomas. The positive correlation between MDCT absolute wash out and ^99m^Tc-HYNIC-TOC is very high, confirming that dual imaging methods have the highest diagnostic contribution to AT’s existence. This study emphasizes the need for upgrading the role of CT imaging in the current diagnostic pathway.

## Figures and Tables

**Figure 1 medicina-60-00072-f001:**
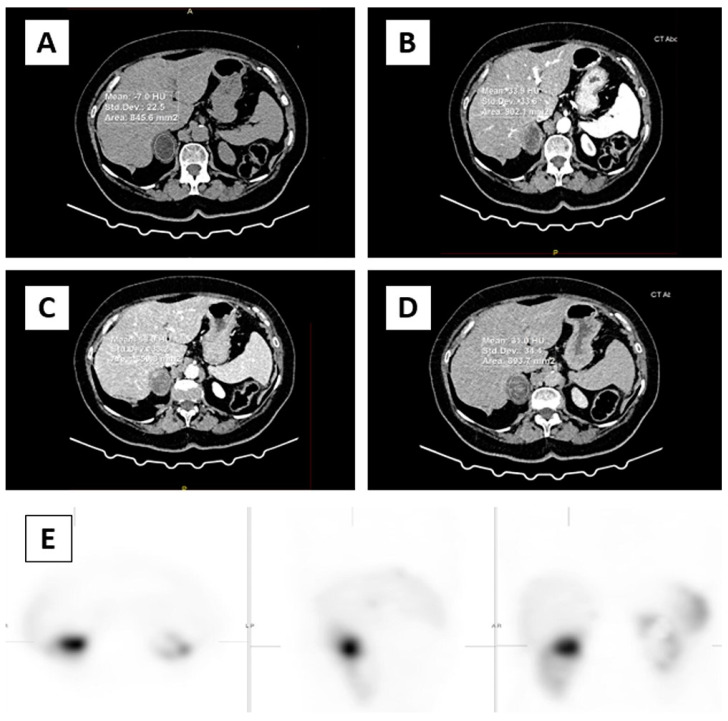
In the axial plane, CT density values (HU units) were measured in all four phases. ROI was placed (circle) over two-thirds of the surface area of adrenal mass in the right adrenal gland (**A**) native phase, (**B**) arterial phase, (**C**) venous phase, (**D**) delayed phase, (**E**) focal intense uptake of ^99m^Tc-HYNIC-TOC in the right adrenal gland correspond to adrenal mass.

**Table 1 medicina-60-00072-t001:** Results of the Wilcoxon Signed Ranks Test between the control and tumor group.

Wilcoxon Signed Ranks Test	Z	*p*
Control group native phase HU—tumor group native HU	−2.462	0.014
Control group arterial phase HU—tumor group arterial HU	−1.045	0.296
Control group venous phase HU—tumor group venous HU	−2.800	0.005
Control group delayed phase HU—tumor group delayed HU	−3.664	0.000
^99m^Tc-HYNIC-TOC target/non-target ratio	−6.154	0.000

**Table 2 medicina-60-00072-t002:** The attenuation HU values of the adrenal masses on native phase and enhanced phases (arterial, venous, and delayed) CT imaging, absolute and relative washout, and ^99m^Tc-HYNIC-TOC target/non-target ratio.

MDCT Examination Phase		N	Mean	SD	Med	Min	Maks
Native phase HU	nonfunctional	29	12.29	17.70	7.00	−13.00	44.00
	secretory	21	11.63	19.96	11.00	−17.00	50.00
	Total	50	12.04	18.39	9.50	−17.00	50.00
Arterial phase HU	nonfunctional	29	74.26	30.80	70.00	29.00	131.00
	secretory	21	75.37	32.12	69.00	30.00	126.00
	Total	50	74.68	30.98	69.50	29.00	131.00
Venous phase HU	nonfunctional	29	85.9	26.77	79.00	40.00	148.00
	secretory	21	89.74	25.12	92.00	35.00	122.00
	Total	50	87.36	25.97	83.50	35.00	148.00
Delayed phase HU	nonfunctional	29	51.03	18.23	50.00	21.00	91.00
	secretory	21	41.05	15.93	37.00	17.00	72.00
	Total	50	47.24	17.90	43.00	17.00	91.00
Relative washout	nonfunctional	29	41.29%	8.97%	42.00%	20.00%	60.00%
	secretory	21	52.89%	10.43%	51.00%	34.00%	68.00%
	Total	50	45.70%	11.03%	44.50%	20.00%	68.00%
Absolute washout	nonfunctional	29	47.77%	7.18%	48.00%	31.00%	59.00%
	secretory	21	64.32%	6.69%	62.00%	56.00%	82.00%
	Total	50	54.06%	10.67%	55.00%	31.00%	82.00%
^99m^Tc-HYNIC-TOC target/non-target ratio	nonfunctional	29	15.9	6.39	15.30	5.99	29.66
	secretory	21	23.76	15.71	17.05	9.69	62.26
	Total	50	18.89	11.42	16.27	5.99	62.26

**Table 3 medicina-60-00072-t003:** Results of the Mann–Whitney U-test for adrenal tumors regarding functional activity.

MDCT Examination Phase	Mann–Whitney U	Z	*p*
Native phase HU	286.500	−0.160	0.873
Arterial phase HU	283.500	−0.220	0.826
Venous phase HU	254. 000	−0.810	0.418
Delayed phase HU	197.500	−1.941	0.052
Relative washout	122.500	−3.446	0.001
Absolute washout	5.000	−5.794	0.000
^99m^Tc-HYNIC-TOC target/non-target ratio	202.000	−1.849	0.064

**Table 4 medicina-60-00072-t004:** Descriptive statistics of adrenocortical adenomas regarding functional activity.

MDCT Examination Phase		N	Mean	SD	Med	Min	Max
Native HU	nonfunctional	29	13.24	17.89	10.00	−13.00	44.00
	secretory	15	10.4	19.42	9.00	−17.00	50.00
	Total	44	12.27	18.25	9.50	−17.00	50.00
Arterial HU	nonfunctional	29	73. 00	31.09	67.00	29.00	131.00
	secretory	15	73.93	32.63	67.00	30.00	126.00
	Total	44	73.32	31.24	67.00	29.00	131.00
Venous HU	nonfunctional	29	85.97	27.66	76.00	40.00	148.00
	secretory	15	84.47	24.51	84.00	35.00	120.00
	Total	44	85.45	26.35	80.00	35.00	148.00
Delayed HU	nonfunctional	29	51.28	18.8	50.00	21.00	91.00
	secretory	15	38.2	14.29	35.00	17.00	70.00
	Total	44	46.82	18.33	43.00	17.00	91.00
Relative washout	nonfunctional	29	41.14%	9.25%	42.00%	20.00%	60.00%
	secretory	15	53.47%	10.34%	52.00%	34.00%	68.00%
	Total	44	45.34%	11.20%	44.50%	20.00%	68.00%
Absolute washout	nonfunctional	29	48.07%	7.27%	48.00%	31.00%	59.00%
	secretory	15	64.80%	7.35%	62.00%	56.00%	82.00%
	Total	44	53.77%	10.79%	54.50%	31.00%	82.00%
^99m^Tc-HYNIC-TOC target/non-target	nonfunctional	29	16.06	6.55	15.30	5.99	29.66
	secretory	15	25.7	16.89	17.05	9.85	62.26
	Total	44	19.34	11.92	16.57	5.99	62.26

**Table 5 medicina-60-00072-t005:** Results of the Mann–Whitney U-test for adrenocortical adenomas regarding functional activity.

MDCT Examination Phase	Mann–Whitney U	*p*
Native phase HU	196.000	0.594
Arterial phase HU	210.500	0.862
Venous phase HU	215.500	0.960
Delayed phase HU	128.500	0.027
Relative washout	81.500	0.001
Absolute washout	5.000	0.000
^99m^Tc-HYNIC-TOC target/non-target	137.000	0.046

## Data Availability

Data are contained within the article and Appendix A.

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
