# Peer review of "Can MDCT Enhancement Patterns Be Helpful in Differentiating Secretory from Non-Functional Adrenal Adenoma?"

_medicina, 2023, doi:10.3390/medicina60010072_

Round 1

Reviewer 1 Report

Comments and Suggestions for Authors

Here are 3 suggestions to strengthen this paper:

1. Expand the discussion on clinical implications and utility of findings. Elaborate more on how these imaging parameters could guide diagnosis and management of adrenal tumors in practice. 

2. Analyze if washout thresholds should be adjusted based on tumor size. Larger adenomas may require customized cutoffs for accurate characterization. 

3. Supplement with pathological correlation after adrenalectomy. This would validate reliability of radiological criteria proposed to distinguish secretory and nonfunctional adenomas.  

Author Response

Dear reviewer,

Thank you very much for taking the time to review this manuscript. Please find the responses in the attacment  and the corresponding revisions/corrections highlighted/ in the re-submitted manuscript

Reviewer 2 Report

Comments and Suggestions for Authors

Manuscript title "Can MDCT enhancement patterns be helpful in differentiating secretory from non-functional adrenal adenoma?"

1. The manuscript aims to answer whether there is a computed tomography (CT) finding helpful in differential diagnosis of functioning adrenal adenoma.

2. The topic may be considered relevant to the field if Introduction and Discussion sections are improved.

3. The main study strength is availability of radionuclide scan and extensive statistical results. Study weaknesses and reviewer's suggestions are outlined below.

4. In the reviewer's opinion, due to lack of clear study hypothesis and gaps in methodology, the conclusions are not consistent with the evidence provided. Additionally, the topic of artificial intelligence is not explored despite the selection of "AI in Imaging—New Perspectives" special issue.

5. The figures are informative, consider adding a study flowchart; the tables are informative.

Specific comments:

- in Abstract, "semi-quantitative analysis" for CT is mentioned; consider adding the findings to "results" section;

- in Introduction, consider adding relevant information regarding ACR adrenal incidental finding management guidelines, the role of magnetic resonance imaging as well as the specific study hypothesis for a more balanced section; what is the rationale behind the presented manuscript given the current diagnostic pathway?

- in Materials and methods, consider providing information on whether study was prospective or retrospective, the reference standard ("confirmed diagnosis of AT" and "secretory adrenal masses"), measurement methodology in the control group (i.e., the HU valued may be inconsistent due to small ROI size); revising contradictory sentences (i.e., "presence of fat, calcifications, septa, necrosis, and cystic degeneration" vs "cystic, necrotic, calcified, or hemorrhagic areas were not included in the measurement"); supplying statements on inter-reader agreement ("there was no inter-reader disagreement"), data normality assessment and sample size estimation

- in Results, consider clarifying what "NT" means, as this acronym is encountered only here; ANOVA appears in this section, despite not mentioned in Materials and methods

- in Discussion, consider summarizing the chief finding, "take home point" in the first sentence, revising contradictory sentences (i.e., "we focused on ... may be used to differentiate and detect subclinical and clinical secretory adenomas in relation to standard laboratory findings" - no such analysis was performed), providing information on other imaging modalities and their shortcomings (as it is unlikely that only CT and SPECT are used for adrenal lesions), and expanding the paragraph on study limitations

Author Response

Dear reviewer,

Thank you very much for taking the time to review this manuscript. Please find the responses in the attachment  and the corresponding revisions/corrections highlighted in the re-submitted manuscript

Round 2

Reviewer 2 Report

Comments and Suggestions for Authors

The authors have provided ample responses to the reviewer's comments. The manuscript's quality has been improved. Additional text editing to correct typos and enhance readability is advisable.